# Open Sensor Manager for IIoT

**Riku Ala-Laurinaho \*** **, Juuso Autiosalo** and **Kari Tammi**

Department of Mechanical Engineering, Aalto University, Otakaari 4 (K1), 02150 Espoo, Finland;
juuso.autiosalo@aalto.fi (J.A.); kari.tammi@aalto.fi (K.T.)
\* Correspondence: riku.ala-laurinaho@aalto.fi

**Abstract:** Data collection in an industrial environment enables several benefits: processes and machinery can be monitored; the performance can be optimized; and the machinery can be proactively maintained. To collect data from machines or production lines, numerous sensors are required, which necessitates a management system. The management of constrained IoT devices such as sensor nodes is extensively studied. However, the previous studies focused only on the remote software updating or configuration of sensor nodes. This paper presents a holistic Open Sensor Manager (OSEMA), which addresses also generating software for different sensor models based on the configuration. In addition, it offers a user-friendly web interface, as well as a REST API (Representational State Transfer Application Programming Interface) for the management. The manager is built with the Django web framework, and sensor nodes rely on ESP32-based microcontrollers. OSEMA enables secure remote software updates of sensor nodes via encryption and hash-based message authentication code. The collected data can be transmitted using the Hypertext Transfer Protocol (HTTP) and Message Queuing Telemetry Transport (MQTT). The use of OSEMA is demonstrated in an industrial domain with applications estimating the usage roughness of an overhead crane and tracking its location. OSEMA enables retrofitting different sensors to existing machinery and processes, allowing additional data collection.

**Keywords:** sensor management; device administration; IIoT; software updates; configuration

## 1. Introduction

Managing sensors has traditionally been a time consuming manual task that has been feasible due to the low update frequency required by the closed nature of legacy systems. Currently, sensors are parts of larger systems such as production lines and often connected straight to the Internet, enabling and necessitating frequent update cycles and opening up new development opportunities. In addition, the number of sensors has rapidly increased as the cost of sensing has been decreased, which further creates the need for efficient management.

Numerous connected devices, such as smartphones, personal computers, and electric cars, are already updated and configured as fleets by their manufacturers to ensure security and deliver the latest improvements. The industry has taken the Internet-connected devices cautiously into use compared to the consumer market, yet due to the advantages of these devices, they are becoming more widely adopted also in industrial environments [1].

The Industrial Internet of Things (IIoT) offers the possibility to collect data from industrial machinery and processes at a low cost. Data collection allows manufacturers to improve manufacturing processes, as well as products by tracking their performance in the field, improve the design and operation, and perform predictive maintenance. For example, the manufacturer of an overhead crane can compare the design parameters and the actual use of the crane and adjust these parameters accordingly for the next generation of products.

A large number of sensors is often a requirement for achieving the projected benefits. Manually adding, configuring, and updating the sensors induce labor costs that can be minimized with a remote management system. Several papers examined the reconfiguration of sensor nodes, including modifying measurement settings or network parameters [2–5]. In addition, a few papers investigated the software update process itself [6–9]. However, the scientific literature regarding a holistic management system and the generation of software based on the sensor configuration and sensor model is scarce. The current solutions do not allow comprehensive configuration changes without storing extra code containing all necessary versions of the functions on the sensor node.

This paper presents the Open Sensor Manager (OSEMA) for wireless sensor nodes, enabling the configuration of the nodes by remote software updates and assisting in sensor node setup via code generation. The source code of OSEMA is publicly available at GitHub [10]. The sensor nodes fetch software updates from the manager and send the measurement data to external data servers. The overview of the measurement setup is illustrated in Figure 1. The paper builds upon the sensor manager presented by the corresponding author in his Master's thesis [11]. The main contributions of this paper are:

- Developing a sensor manager that allows remote software updates of sensor nodes via a web user interface and a REST API (Representational State Transfer Application Programming Interface);
- Introducing a software generation method for sensor nodes with minimal manual programming;
- Implementing the configuration of sensor nodes via remote software updates, which allows changing sensor settings via the I2C (Inter-Integrated Circuit) bus.

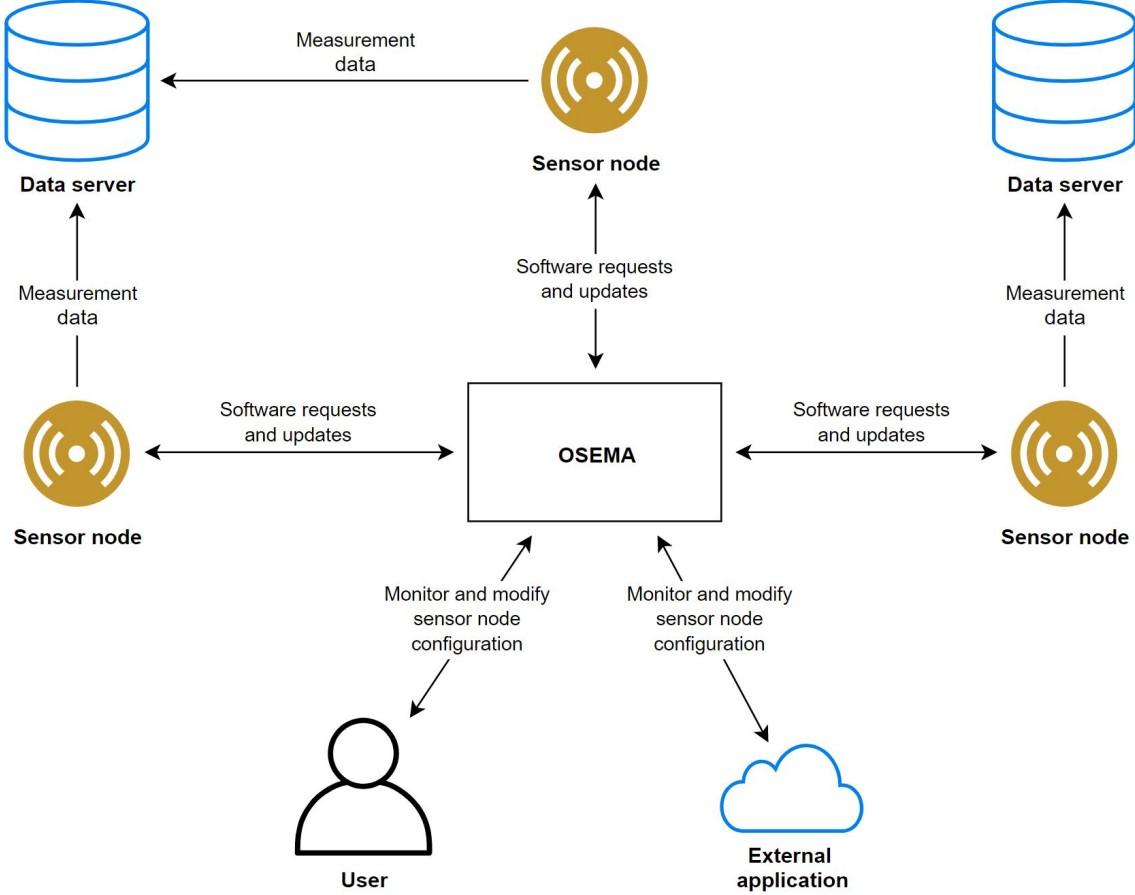

**Figure 1.** An overview of data collection with the Open Sensor Manager (OSEMA).

## 2. Related Work

Several papers examined the configuration of the constrained devices, including modifying the measurement settings and network parameters of the device. An approach based on the

Arrowhead Framework for the deployment and management of IoT (Internet of Things) devices was presented in ref. [2]. The proposed approach aids the creation of cloud-based automation systems. Configuration methods adopting a smart gateway were also proposed [3,4]. The first proposal [3] adopted Constrained RESTful Environments (CoRE) Link Format and Sensor Markup Language (SenML) to describe the available sensors, actuators, and the metadata. APIs were then used to modify these configuration data. In ref. [4], the configuration was performed by modifying the configuration file of the device, which was in XML format. The configuration was also possible directly without the gateway. The configuration of Bluetooth Low Energy beacons without Internet connectivity in smart cities was demonstrated in ref. [5].

The software and firmware of devices are updated to add new features and functionalities or to fix bugs and remove vulnerabilities. This paper does not differentiate remote updating of software and firmware, as typically, similar methods can be used to conduct them. Updating the firmware of devices remotely is called Firmware Over-The-Air (FOTA). FOTA is commonly employed with mobile devices such as smartphones or tablets [12]. However, performing firmware updates for constrained devices calls for a standardized method.

To address firmware updates and device management, the Open Mobile Alliance (OMA) has standardized OMA Device Management (DM) for mobile devices (e.g., smartphones) [13] and Lightweight Machine-to-Machine (LwM2M) for constrained devices [12]. LwM2M is a lighter version of OMA DM [14] and offers several features for the management, such as monitoring, updating firmware, and the configuration of the devices [6]. In the standardized OMA LwM2M architecture, the device must support CoAP (Constrained Application Protocol) to implement the LwM2M Firmware Update object, but the firmware update itself can be downloaded using other protocols such as HTTP (Hypertext Transfer Protocol) [15]. In ref. [6], the LwM2M architecture, which offered also an interface for FOTA, was implemented for the Texas Instruments CC2538 System-on-Chip running the Contiki operating system.

Several non-standardized architectures and methods for conducting remote software updates have been presented. A solution based on Generic extension for Internet-of-Things Architectures (GITAR) [7] allows partial network-level software updates on constrained devices. The partial software updates reduce the memory needed to perform updates compared to updating the whole software. Updating the firmware of Espressif ESP8266 over HTTP and Wi-Fi was demonstrated in ref. [8]. Requirements for the device management were identified, and a high-level framework for the management was introduced by Mohapatra et al. [16].

Several closed-source [17–20] and open-source [21,22] IoT platforms with various support for FOTA and the configuration of devices are also available. Amazon Web Services (AWS) allows Over-The-Air (OTA) updates for both AWSIoT Greengrass and FreeRTOSdevices [23]. With FreeRTOS devices, updates can be sent over HTTP or MQTT [24]. In addition, any file can be sent using the OTA feature, which could be used for example to change configuration files. Microsoft Azure IoT also allows remote firmware updates and offers automatic device management [25]. The IBM Watson IoT platform implements device management via the Device Management Protocol, which is based on MQTT [26]. Device management includes remote firmware updating, which is implemented by providing the necessary information such as the download URI (Uniform Resource Identifier) for the devices to perform the update [27].

Kaa IoT was initially developed as an open-source project, but has since been moved to a closed-source repository and branded as an enterprise cloud IoT platform [28]. It offers over-the-air orchestrator service, which provides necessary information for performing remote software updates [29]. DeviceHive is an open-source IoT platform, which supports HTTP, Websockets, and MQTT for communication [30]. It is possible to send commands to devices, which could be used to change their configuration. Thinger.io is an open-source IoT platform, which is scalable and hardware agnostic [31]. Input and output resources can be defined for each device, which allows interaction with a device via REST API [32].

　　　　Security is a crucial part of software updating: without properly implemented security, an attacker may modify the update, disabling the device or making it produce false sensor readings. To improve security, a secure FOTA object based on a secure JSON (JavaScript Object Notation) object was proposed [12]. A secure over-the-air programming framework, which relies on symmetric encryption with AES (Advanced Encryption Standard) 128bit CBC (Cipher Block Chaining), was presented in ref. [9]. The framework is open-source and offers integrity, confidentiality, and authentication. The current implementation of the framework supports ATmega2560-based IoT devices such as Arduino.

　　　　A comprehensive review of software updates over-the-air on wireless sensor networks was presented in ref. [33]. The review investigated protocols for disseminating the update and reducing the network traffic. An efficient updating procedure of code in wireless sensor networks based on updating only the necessary parts of the code was presented in ref. [34]. The developed system adopted the "Lightweight Mesh (LWMesh) network protocol over Peer-to-Peer Mesh (P2PMesh) architecture" and offered low energy consumption. A system for updating firmware over-the-air in mesh networks was introduced in ref. [35].

　　　　As a conclusion, an extensive amount of research has been conducted on the configuration and remote updating of the software of constrained devices. However, studies examining the ways in which new sensor models can be added to management systems or software can be generated based on the configuration of the sensor node are scarce. Current solutions for the configuration of sensor nodes require that a sensor node contains the code for all possible configurations because only configuration parameters are modified. This applies also to platforms, in which commands can be sent to devices. Remote software updates can be used to change a sensor node configuration; yet, it requires code generation based on the sensor configuration. This feature could not be identified in any of the reviewed papers or solutions.

## 3. Materials And Methods

### 3.1. Sensor Node

　　　　A sensor node consisted of a microcontroller and a sensor (Figure 2a). The microcontrollers used were based on the Espressif ESP32 chipset, and they had built-in Wi-Fi and Bluetooth. The microcontrollers were selected from PyCom, because they offered boards also with built-in NB-IoT (Narrowband IoT), LoRa (Long Range), and Sigfox. This would allow implementing support for several different networks in the future.

　　　　Sensors must have an I2C bus, which is a two-wire communication bus with low energy consumption [36]. Currently, OSEMA supports two three-axis accelerometers, namely ADXL345 [37] and LIS3DSH [38], a LiDAR (Light Detection and Ranging) distance sensor (Figure 2b, LIDAR-Lite v3 [39]), and a 12 bit analog-to-digital converter.

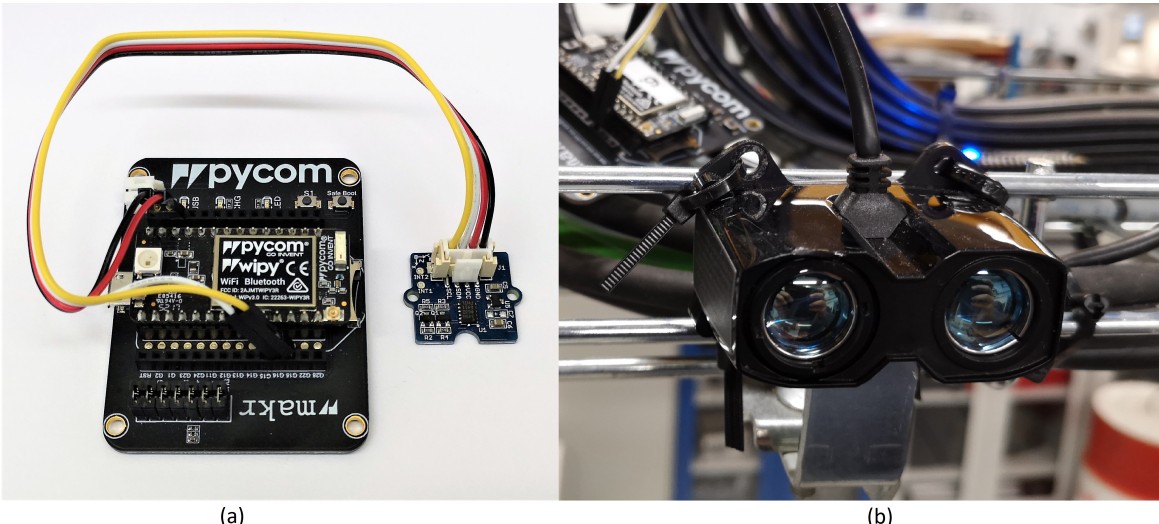

(a)                                                                    (b)

**Figure 2.** (**a**) A sensor node consists of a microcontroller and a sensor such as a three-axis accelerometer. (**b**) A LiDAR sensor attached to the bridge of an overhead crane was used to track the location of the crane.

### 3.2. Programming, Frameworks and Tools

The Django web framework was adopted in the development of OSEMA. Django is an open-source Python web framework that allows rapid development of web applications [40]. It offers several built-in features such as user authentication and administration. The features of Django can also be extended with numerous add-ons, of which the following were employed in the development of the system: the Django REST framework [41] and the REST Framework Generic relations [42] for creating the REST API and Simple JWT (JSON Web Token) [43] for token-based authentication. A responsive and clear Web User Interface was created with Bootstrap [44] and jQuery [45], which was applied for manipulating documents, handling events, and asynchronous communication with a back-end. OSEMA was deployed with the Apache server on Raspberry Pi 3 Model B+.

The programming of microcontrollers relied on MicroPython, which is a Python 3-based programming language developed for microcontrollers and constrained environments [46]. It is small in size and can be used with only 8 kB of RAM [47]. MicroPython has automatic memory management and can be considered an easy language to program with for those familiar with Python 3.

## 4. Open Sensor Manager

OSEMA allows the management of sensor nodes remotely over the Internet and the quick setup of the nodes via software generation. To collect data, the sensor nodes and data server(s) are also needed, as shown in Figure 3. The configurations of sensor nodes are stored in the database of OSEMA and can be modified via the REST API and web user interface. If a node configuration is modified, OSEMA generates a new software based on the new configuration. Sensor nodes request these software updates from OSEMA periodically over HTTP. The measurement data are sent over HTTP or MQTT (Message Queuing Telemetry Transport) to an external data server.

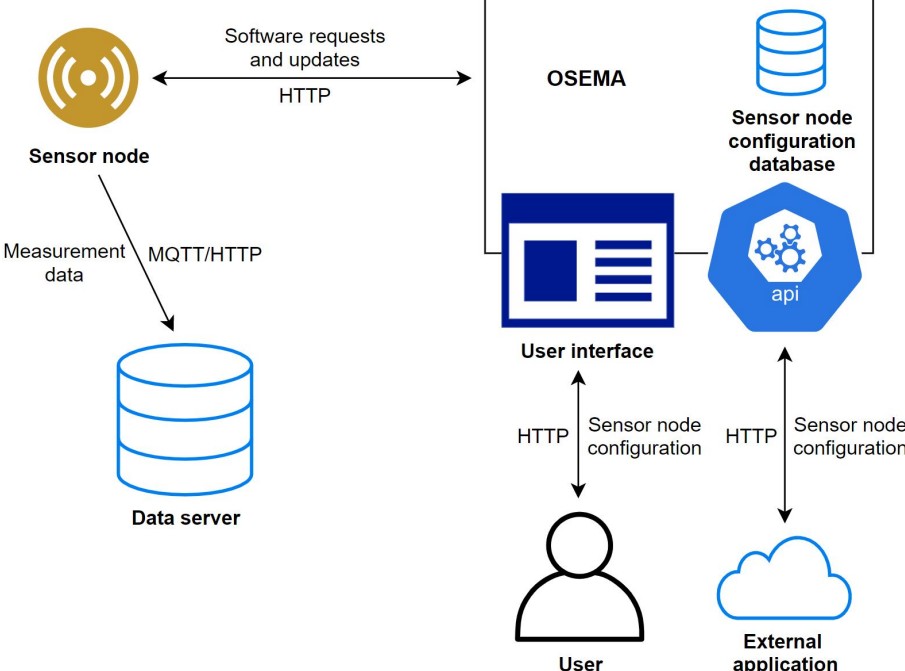

**Figure 3.** Detailed overview of a measurement setup.

### 4.1. Generation of Software for a Sensor Node

After sensor node has been added or modified in OSEMA, OSEMA generates the software based on its configuration. The configuration consists of a sensor model, measurement and data transmission settings, network parameters, the application layer protocol, and the data format. A complete list of the configurable settings is shown in Table 1. The configurations of nodes are stored in the database of OSEMA. The node software is generated by conditional statements, which select the correct versions of the functions and libraries based on the configuration. For example, if data are to be transmitted via MQTT, the corresponding library is written to the software. In general, the software generation based on the configuration allows the software to contain only necessary parts, which reduces its size. This enables the high customizability of the software with no additional demands of the computational power or memory size of a microcontroller.

**Table 1.** Configurable settings of sensor nodes (updated version from [11]).

| Setting | Description | Example Value |
|---|---|---|
| Update URL | OSEMA server URL where updates are requested | www.example.com |
| Update port | Sensor node requests updates from this port | 443 |
| Update over HTTPS | Use HTTPS for sending updates (true/false) | True |
| Update check limit | How often a node requests updates | 3600 (s) |
| Sensor model | Which sensor model is used | ADXL345 |
| Sample rate | Sample rate of the sensor | 12.5 (Hz) |
| Sensitivity | Sensitivity of sensor | $\pm 2$ g |
| Burst length | Length of measurement when measured in bursts | 10 (s) |
| Burst rate | Time interval between bursts | 10 (s) |
| Data send rate | How often data are sent to the data server | 10 (s) |
| Connection close limit | If the data send rate are higher than this, the connection is closed after data transmission | 3 (s) |
| Network close limit | If the data send rate are higher than this, the network connection is closed after data transmission | 30 (s) |
| SSID | SSID of the Wi-Fi network | myHotspot |
| Security | The security method of the Wi-Fi network | WPA2 |
| Key | Password of the Wi-Fi network | secretpasswd |

**Table 1.** *Cont.*

| Setting | Description | Example Value |
|---------|-------------|---------------|
| Username | Username for Wi-Fi network | username |
| Protocol | Application layer protocols used for measurement data transmission | MQTT |
| Data server URL | Data server URL | www.example.com |
| Data server port | Data server port | 80 |
| Path | Path where the measurement data are sent (with HTTP/HTTPS) | /add-data |
| MQTT User | Username for MQTT | username |
| MQTT Key | Password for MQTT | secretpasswd |
| Topic | MQTT topic | example/topic |
| Broker URL | MQTT broker URL | io.adafruit.com |
| Broker port | MQTT broker port | 1883 |
| Data format | In which format measurement data are sent | JSON |
| Variable name | Each variable should be named | x_acceleration |
| Encrypt data | Encrypt the measurement data (true/false) | True |
| Key for data encryption | Key used to encrypt data as hex number | 3ba19f5c4192b131 7123e993ca9dae21 |

For each sensor model, the available sample rates and sensitivities need to be separately added to OSEMA. This is conducted by adding register-value pairs to the manager, which then writes them to the generated software based on the sensor node configuration. The values are written to the corresponding registers via the I2C bus by the node in the boot-up. This design allows the use of different sensors without sensor-specific libraries. Thus, modifications to the software generation code were seldom needed, when adding a new sensor model. In addition, function handling raw sensor data could be added for each sensor model, which would allow converting measurement data to numeric values and preprocessing of the data.

*4.2. Software Update Process*

The software update process and how user action triggers a software update are illustrated on the right side of Figure 4. A sensor node periodically asks for updates from the management system with an HTTP request containing the current software version of the node. OSEMA compares this version number to the newest software version in its database and either responds with new software in the payload of the response or confirms that the sensor node has the latest software version. If new software is received, the node writes it to its flash memory and reboots. In the boot-up, the node replaces the old software version with the new one.

*4.3. Measuring*

The microcontroller reads measurement data from the sensor via the I2C bus at the user-defined sample rate. The data are stored in the flash memory of the microcontroller and sent periodically to a data server. For sending the data, a new thread is created, which allows continuing the measurement during data transmission. The new thread formats the data, possibly encrypts it, and sends it over HTTP, HTTPS (HTTP Secure), or MQTT to the data server. The measuring algorithm of a sensor node is described on the left side of Figure 4. The software updates and data collection are run on separate threads on the microcontroller. In addition, one thread takes care of the network connection.

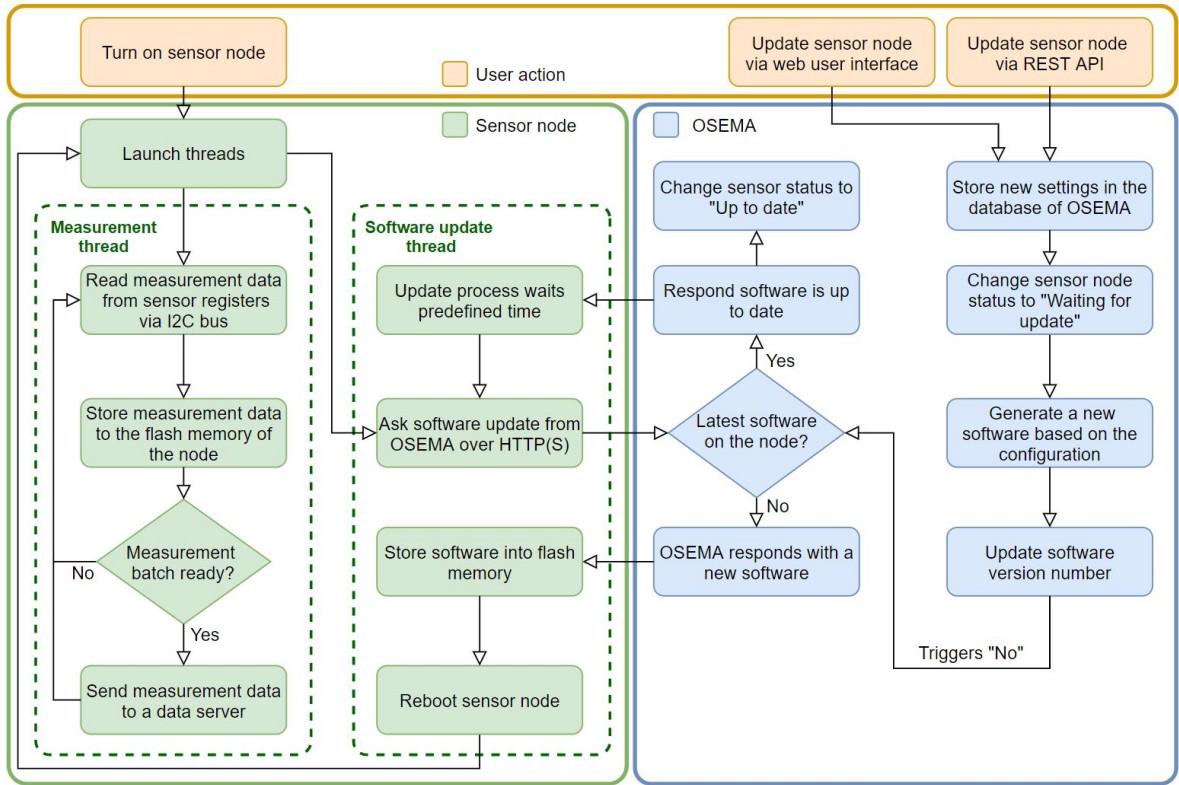

**Figure 4.** The operation of a sensor node and its interaction with OSEMA.

## 4.4. Security

OSEMA was aimed to be used in an industrial environment, and therefore, the requirements for the security of the manager were high. Companies want to prevent information leaks to competitors and other untrusted parties. This information includes, for example, the activity of the factory, which data are collected from machines, and the measurement data. Therefore, the integrity of the software updates is not enough: by analyzing the updates, information about the process and machines could be leaked. Therefore, the confidentiality of firmware updates needs to be ensured. The security of firmware updates was also suggested in ref. [48], which presented a more comprehensive analysis of the threats to and security of IoT.

The security model was based on ensuring confidentiality with encryption, as well as authenticity and integrity with message authentication code. Requests and responses for software updates are always encrypted with AES 128-bit CBC, which is considered a secure encryption method [49]. For each message, a keyed-Hash Message Authentication Code (HMAC) is calculated with SHA-256 (Secure Hash Algorithm) and a 128 bit key to ensure the authenticity and integrity of the message. Measurement data can also be secured using a similar method if the data format is JSON. The shared secret keys are created for the sensor nodes at the time of software generation. The keys are stored in the flash memory of the microcontroller unencrypted. Thus, if an attacker gains physical access to the microcontroller, the keys can be stolen. However, OSEMA is designed to be used in closed industrial environments, and if an intruder can access the factory floor, the security of the factory is already seriously compromised.

In addition to encryption and HMAC, a nonce is used to prevent replay attacks with software updates. The nonce is a randomly-generated 128 bit key, which is sent along with a software request. The response contains the same key, and the node checks if it matches the one sent with the request. By replaying an old software update, the configuration of the node could be changed, which could cause a sensor to produce inadequate data or, in the worst case, even stop the measurement. In addition, software updates could be prevented by replaying the "software up to date" response. A replay attack

is possible if, for example, at some point on the network path, an attacker can listen to the traffic or the security of Wi-Fi is compromised.

The software update process can be performed over HTTP or HTTPS. Because the support for certificate checking of the used microcontrollers is restricted, a man-in-the-middle attack is possible even with HTTPS. Therefore, HTTPS alone does not provide sufficient security, and the aforementioned encryption scheme was used. The symmetric encryption with AES 128 bit is a relatively efficient method for encryption [9] and provides a high level of security regardless of the application layer protocol.

### 4.5. Application Layer Protocols

The measurement data from sensor nodes can be sent over HTTP, HTTPS, or MQTT to the data server. MQTT is a publish/subscribe protocol running on top of TCP [50]. The protocol has a bit formatted header inducing a small overhead. In addition, it has low energy consumption, and its CPU and memory usage are low [51]. Therefore, it is suitable for constrained devices such as sensor nodes.

HTTP is a widely used request-response protocol, which runs on top of a reliable transport layer protocol, TCP (Transmission Control Protocol) [52]. It has text-based headers [53] leading to a relatively large overhead size. In addition, HTTP has high energy consumption, bandwidth usage, and requirements for computational power [54]. Therefore, HTTP is not especially suitable for constrained devices. However, the support for sending measurement data over HTTP was implemented due to its ubiquity, which enabled compatibility with other services. HTTPS offers secure communication by sending data over TLS (Transport Layer Security) [55].

### 4.6. Web User Interface

OSEMA has a web user interface, which allows managing sensor nodes. Figure 5 shows the browse sensors view, which lists all sensor nodes in the manager. In this view, the statuses of sensor nodes can be monitored, and their configuration can be changed by selecting the edit button. When a sensor node is added (Figure 6), it is configured according to the list presented in Table 1. In addition, a description and location of the node can be defined.

**Figure 5.** Browse sensor view, which shows all sensor nodes in OSEMA.

## Add sensor

* = Required

### General settings

Sensor name:*

Description:

Location:

Update url:*

Update port:*

Update over HTTPS :

### Measurement settings

Model:*

ADXL345

Sample rate (Hz):

3.13

**Figure 6.** A part of the add sensor view.

## 5. Use Cases of OSEMA

OSEMA supports multiple sensor models, application layer protocols, and data formats to enable interoperability, which is highly recommended in a multi-sensor context [56]. In addition, it offers a REST API, which further extends the versatility of the manager. Therefore, OSEMA can be employed in various use cases and applications, three of which are presented below. The system is especially suitable for use cases in which a large number of different types of sensors are needed and/or the measurement settings are regularly changed.

### 5.1. Usage Roughness of an Overhead Crane

The driving style of an overhead crane affects the lifetime of the components. For example, compared to moderate accelerations, rapid ones yield larger forces that wear components faster. A rough driving style also increases the risk of accidents. Thus, a system for estimating the usage roughness of an industrial overhead crane, Ilmatar [57] (Figure 7), was developed. The movement of the hook was measured with a three-axis accelerometer (Figure 8), which sent sensor readings constantly to an externally managed IoT platform via MQTT. The platform then calculated the usage roughness index from the measurement data using a machine learning algorithm, which was also implemented by the same external party. The operator of the crane received feedback on the driving style with a simple gauge. Figure 9 shows the interaction between entities in the usage roughness estimation application.

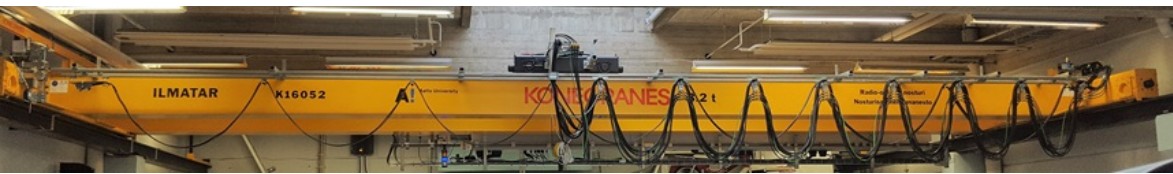

**Figure 7.** An overhead crane located at the Aalto Industrial Internet Campus [11].

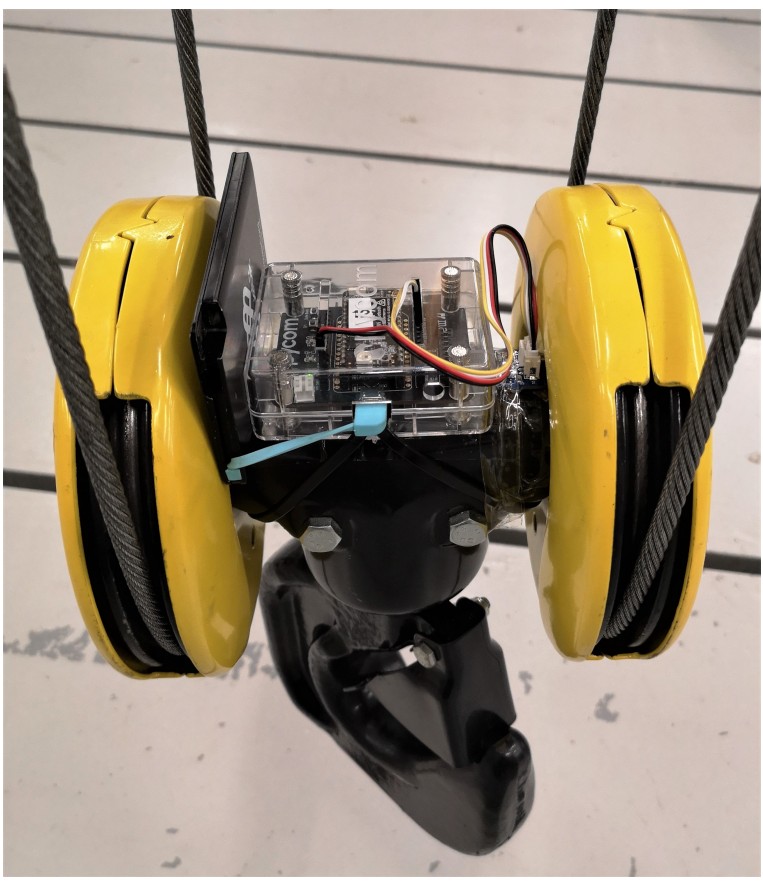

**Figure 8.** A three-axis accelerometer was attached to the hook of an overhead crane to estimate the usage roughness of the crane.

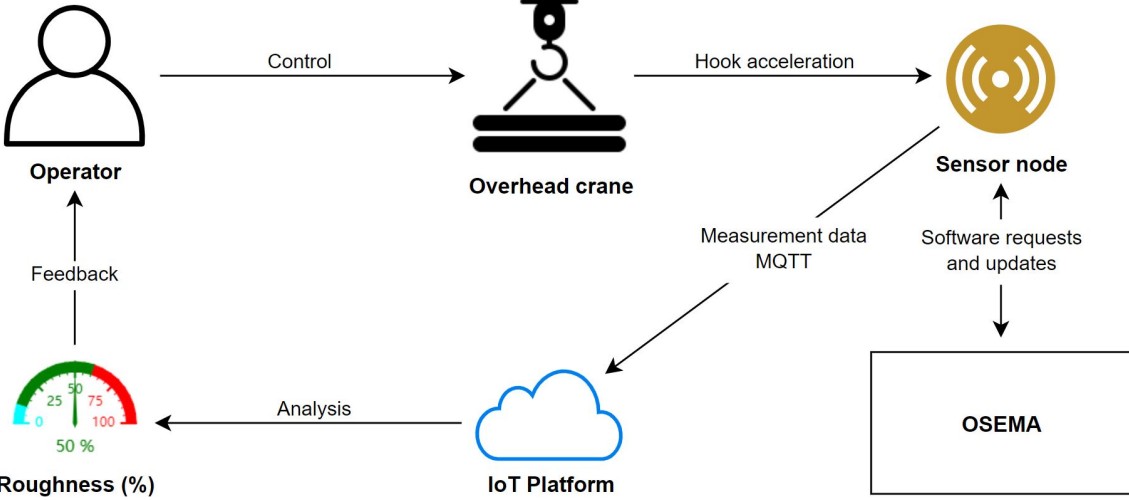

**Figure 9.** The system overview of the usage roughness estimation application. The operator controls the crane; a sensor measures the acceleration of the hook; and an IoT platform performs analysis based on the data and gives feedback about the driving style.

The application was validated by operating the overhead crane with different driving styles. The tests indicated that even a simple measurement setup was able to distinguish rough driving from the normal use of the crane. However, modern cranes have built-in anti-sway control and automatically limit maximum acceleration, which reduces the usefulness of the application. These features already restrict the roughness of the driving and must be turned off to yield a high roughness index. The use case showed that OSEMA allowed rapid application development even remotely. The configuration

of the node could be changed easily and would be possible even by the algorithm developer if the measurement data were inadequate. Compared to other systems, setting up the measurement was swift and easy, since the three-axis accelerometer was already configured to OSEMA, and manual programming was not needed. In addition, changing the sensor measurement settings was quick, which allowed experimenting with various sample rates.

## 5.2. Position Tracking of an Overhead Crane

Tracking the position of an overhead crane enables calculating the distance driven with the crane. When combined with load data, the remaining lifetime of the crane and its components can be estimated. In addition, position tracking allows analyzing and optimizing the routes of the crane in a factory. The position of the crane was tracked with two LiDAR distance sensors (Figure 2b), one of which measured the position of the bridge (x-axis) and the other the position of the trolley (y-axis). The measurement data were transmitted using MQTT to an external IoT platform. With OSEMA, it was easy to test different measurement settings and find a suitable configuration for the use case.

The overhead crane used for testing had built-in laser sensors to measure the position of the crane, which were used as a reference for measuring the accuracy of the position tracking system. The preliminary tests indicated that the accuracy of the LIDAR sensors was sufficient for the position tracking system and to track the distance driven by the crane. However, some tasks, such as automated high precision assembly lift [58], require significantly more accurate sensors.

## 5.3. Laboratory Exercise

OSEMA was used as a part of laboratory exercise in a master-level mechatronics course. In the exercise, the students set up a sensor node to investigate the sway of the hook of an overhead crane with a three-axis accelerometer (Figure 8). First, they needed to add a new sensor node to OSEMA, after which the software generated by OSEMA was uploaded to the microcontroller. Finally, the acceleration of the hook was observed from a specified data server, which plotted the acceleration. The measurement data were sent using HTTP and JSON data format to the data server. All cables were connected, and the necessary software (firmware update tool and FTP client for uploading files) was preinstalled.

A total of 49 students participated in the laboratory exercise, and 42 of them worked in pairs. Thus, the number of responses to the feedback questionnaire was 28. On average, it took 27 min 45 s from starting the exercise for measurement data to appear on the data server. The questionnaire results showed that OSEMA was considered reasonably easy to use (Figure 10): the average rating for the ease of use was 2.11 (1 = very easy, 5 = very difficult). Most problems encountered during the tests were related to moving the files from a laptop to the microcontroller because the files could not be renamed or deleted after they had been transferred to the microcontroller. Therefore, if the wrong file was uploaded to the microcontroller, all files had to be removed using the firmware update tool. This negatively affected the workflow, but could not be fixed with OSEMA, since the problem was caused by the microcontroller firmware.

The laboratory exercise demonstrated the use of OSEMA outside the industrial environment. OSEMA enabled the quick setup of the measurement, which allowed the students to concentrate on observing the phenomenon and the analysis of the data, instead of the programming of microcontrollers. In addition, students could easily try different measurement parameters. For example, the effect of the sample rate on showing the phenomenon could be tested. Compared to other IoT platforms, manual programming was not needed, but data storing and visualization had to be implemented using other services or platforms.

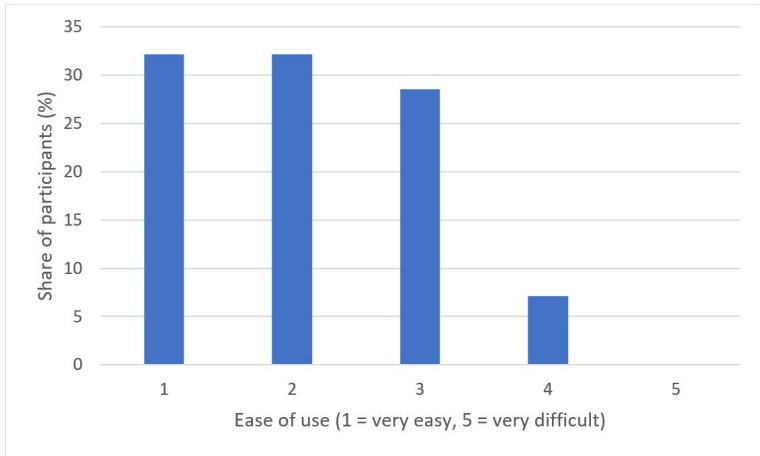

**Figure 10.** The ease of use of the management system assessed by students of a mechatronics course.

## 6. Discussion

There is no standardized method for adding sensors to IoT devices: each sensor has to be configured separately, and manual programming is required. Compared to the other systems, OSEMA minimized manual work: once a new sensor model was added to the manager, deploying a new sensor node was straightforward, since the code could be generated by the platform, and the user needed only to choose the desired configuration. In the future, the users could share the sensor models they have added to the manager with others, further reducing the need for manual programming.

The generated software could be sent to the nodes remotely. Software update stopped the measurement for approximately ten seconds and made configuration changes slower compared to solutions in which only the configuration file is modified. In addition, replacing the whole software required more data to be sent compared to the partial updates presented in ref. [7]. However, because the software was not expected to be updated particularly often (rather once in a month than once in an hour), the extra network usage was not considered as a major drawback. On the other hand, the implementation of partial code updates would have required a larger and more complex code.

One of the main limitations of the manager was support for only ESP32-based microcontrollers programmed with MicroPython and I2C bus sensors. In addition, only HTTP(S) and MQTT could currently be used to send measurement data. This was due to the restricted availability of application layer libraries for MicroPython. However, support for CoAP could be added in the future with a recently published library [59]. Wi-Fi is currently the only supported network, which restricts the number of devices per area.

OSEMA offered high-level security with established encryption schema. Despite the encryption of the messages, an attacker could determine when the software was updated or how often the measurement data were sent to the server by capturing the network traffic. In addition, some information about the update could be obtained, by analyzing the length of the update. This could be prevented with constant-length updates. However, overhead and energy consumption would increase because extra data must be sent.

Several directions for future research have been recognized. The possibility to conduct only partial code updates would reduce the transferred data. The program generation functionality could be also added for the C language, which is widely used with constrained devices. The implementation of other networks is possible, as microcontrollers supporting NB-IoT, LoRa, and Sigfox are available. The usability of the user interface could be further improved by using object detection from a video stream, as presented in ref. [60]. To address the development of the manager, the source code is publicly available on GitHub [10]. The development of OSEMA is planned to continue as an open-source project.

## 7. Conclusions

Data collection from machinery enables improving operation and optimizing maintenance. Due to low-cost sensors and microcontrollers, as well as inexpensive Internet connectivity, data collection has become more feasible. However, as the amount of sensor nodes increases, their management becomes cumbersome. This paper presented an open-sensor manager, OSEMA, which allowed remote configuration and software updates of sensor nodes over the Internet via the graphical user interface and REST API. Compared to the other solutions for the management of the sensor nodes, OSEMA allowed more comprehensive configuration changes with less code stored on the microcontroller and configuration of I2C bus sensors.

The security of the manager was at a high level. Software updates were always sent encrypted with AES 128 bit CBC, and HMAC was calculated with an SHA-256 hashing algorithm. In addition, measurement data could be sent encrypted. Two industrial applications, where the system was employed, were presented. In the first application, the usage roughness of an overhead crane was estimated with a machine learning algorithm, and in the second, the position of the crane was tracked. The use of OSEMA as a part of a mechatronics course indicated that the system was fairly easy to use and that a new sensor node could be added to the system in less than 30 min. The source code of the platform is available on GitHub [10], and the development of the platform continues as an open-source project.

**Author Contributions:** Conceptualization, R.A.-L. and J.A.; methodology, R.A.-L. and J.A.; software, R.A.-L.; validation, R.A.-L., J.A., and K.T.; investigation, R.A.-L.; resources, K.T.; writing, original draft preparation, R.A.-L.; writing, review and editing, J.A. and K.T.; visualization, R.A.-L.; supervision, K.T.; project administration, J.A.; funding acquisition, J.A. and K.T. All authors have read and agreed to the published version of the manuscript.

**Funding:** This work was supported by the Business Finland under Grant 8205/31/2017 "DigiTwin" and Grant 3508/31/2019 "MACHINAIDE".

**Acknowledgments:** The authors would like to thank Ivar Koene for his help with the microcontrollers and Petri Kuosmanen for enabling this study. The authors also acknowledge Miika Valtonen and Mika Hannula for the development of the usage roughness estimation algorithm. The authors would like to thank Atso Galkin for his valuable comments about the visuals and help with writing. Finally, the authors would like to thank Joel Mattila for implementing the data server for the mechatronics course.

**Conflicts of Interest:** The authors declare no conflict of interest.

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
