# Peer review of "Open Sensor Manager for IIoT"

_jsan, doi:10.3390/jsan9020030_

Round 1

Reviewer 1 Report

The authors developed an open sensor manager, OSEMA, which addresses generating software for different sensor models and offers a user-friendly web interface. Overall, this is a well-structured paper that aims to introduce the composition and application of a new sensor manager, with some original propositions made, but it needs some revisions before being accepted.
1. Some abbreviations need to be given a complete expression before when they appear for the first time in the paper. Such as REST API.
2. The Section 6(User tests) seems not to be persuasive enough. You can add a control group using other similar platforms, or just add the test results for users in Section 5.
3. “Most problems encountered during the tests were related to moving the files from a laptop to the microcontroller because the files could not be renamed or deleted after they have been transferred to the microcontroller. Therefore, if a wrong file is uploaded to the icrocontroller , all files have to be removed using the firmware update tool. "Is this where OSEMA needs to be further improved? If not, it may not be necessary for this paragraph to appear in the paper.

4. At least 3 other related newly published(2019&2020) papers from Journal of Sensor and Actuator Networks should be added and cited.

Reviewer 2 Report

The paper presents an interesting holistic open sensor manager, which enables remote configuration and software updates of sensor nodes over the Internet. The solution is well presented and discussed. However, in the presentation of the use cases, the originality of OSEMA is not clear. The authors should show the advantages of the use of their solution with regard to other solutions.

Reviewer 3 Report

This manuscript presented a web-based lightweight IOT manager which can be run on Raspberry Pi. The proposed manager can add, configure, update the IOT devices, and transmit the sensing data. These features are presented as manuscript promised. However, all these function or features seems fully already achieved (see reference) and robustness of high frequency and large scale sensing practice was no show. The proposed method should compare to these reference. Author may more emphasize the novel features and large scale sensing, and resulting device/data dashboard to stand out the contribution.

Ref.
product based:
QIoT Suite
AWS IOT Edge
Azure IoT
IBM Watson IoT Platform

open source based:
Kaa IoT Platform
SiteWhere: Open Platform for the Internet of Things
ThingSpeak: An open IoT platform with MATLAB analytics
DeviceHive: IoT Made Easy
Zetta: API-First Internet of Things Platform
DSA: Open Source Platform & 「Toolkit」 for Internet Of Things Devices
Thinger.io: The Opensource Platform for Internet of things
WSo2- Open source platform for Internet of Things and mobile projects

Reviewer 4 Report

This article introduces a software solution to manage constrained IoT devices such as sensor nodes in IIoT systems and use cases. Overall, the paper is well-structured and easy to read. However, in most of its parts, the article has a strong focus on engineering aspects, whereas the scientific contribution remains vague. This is maybe due to the fact that the article is based on a master thesis.

Still, I believe the article can be interesting for the audience of this journal. Before publication, I suggest the authors to eliminate the following weaknesses.

  • I general, it appears the article wants too much. It surveys related work, it introduces the functionality of the proposed solution, it discusses security aspect, it elaborates on the usability of the proposed solution, it shows two application scenarios, and it revisits all these topics again in a discussion section. While this is ambitious and each aspect is interesting, it leads to the problem, that the article fails to achieve a sufficient level of detail in most of the mentioned aspects. For instance: (1) The article mentions some implemented security measures but fails to introduce a comprehensible security model. (2) The article touches the topic “usability”, but obviously no scientifically sound usability study has been conducted. (3) The article briefly shows 2 concrete use cases, but no detailed evaluation figures are provided. In general, I would prefer the article to focus on less aspects but address those aspects in more detail and in a more scientifically sound way.
  • I in particular had problems to understand the security model behind the proposed solution, especially regarding the provision of IoT devices with software updates. I understood that TLS is applied but cannot be fully trusted, as IoT devices have limitations in validating certificates. Consequently, additional security measures are applied (AES, HMAC, etc.). This raises the following questions that are not answered adequately by the article:
    • Why is encryption of delivered software updates needed at all? I understand that the integrity authenticity of software is important. However, to achieve that, HMAC would suffice. The authors should explain why confidentiality needs to be assured as well.
    • Assuming that delivered software indeed needs to be encrypted: How do the communicating parties (i.e. managing software, IoT devices) exchange the required symmetric encryption keys? How can these keys be stored securely on IoT devices? If you cannot protect the keys adequately, any encryption is useless.
    • What is the attack vector behind the mentioned replay attack? Which message could an attacker replay to gain any advantage? The authors should elaborate on the concrete attack scenarios they aim to avoid. Otherwise, the use of cryptographic algorithms seems arbitrary.
    • How is the mentioned “session key” used to protect against replay attacks? Where is this key derived from? How is it applied? The authors need to provide much more details in order to make the reader understand the underlying security model.
  • The flow chart in Figure 4 is confusing. Starting at the intended “starting point” some states in the chart (i.e. those on the right part) are never reached. Also, transitions between states are sometimes unambiguous. I think it would be better not to try to reflect concurrency in this type of diagram but to choose some other way of illustrating parallel threads.
  • The Discussion section is redundant to a large extent. It mainly repeats what has already been discussed in preceding sections.

Round 2

Reviewer 1 Report

The paper is revised carefully according to the reviewing comments,the acceptance is suggested.

Author Response

Authors would like to thank reviewer for the comment.

Reviewer 4 Report

Overall, the authors did a good job in improving the article and eliminating the weaknesses pointed out in the first review. The article is now more focused, and several security-related open questions are answered. Also, the flow chart in Figure 4 has been improved significantly.

I am still not fully convinced by Section 6 (Discussion), though. This section is rather a conclusions-like summary of the paper than a discussion. However, I do not rate this a blocking issue that prevents publication of the article. Still, I would recommend the authors to improve this section or to delete it and move its contents to other section where appropriate.

A remark to the security section and the implemented replay protection: I suggest referring to the secret “session key” as “nonce”. This is the usual notation regarding replay protection.

There are still some typos that should be removed before publication, e.g.:

  • Line 41: “[…] solutions does […]” should be “[…] solutions do […]”
  • Line 323: “[…] has to configured […]” should be “[…] has to be configured […]”
  • etc.

Overall, the paper is still very engineering-focused, and the scientific contribution is limited. Still, I believe the article could be interesting for a certain target group.
